# Black Rot of Grapes (*Guignardia bidwellii*)—A Comprehensive Overview

Márton Szabó [1], Anna Csikász-Krizsics [1], Terézia Dula [2], Eszter Farkas [3], Dóra Roznik [1], Pál Kozma [1] and Tamás Deák [3,*]

1   Research Institute for Viticulture and Oenology, University of Pécs, Pázmány Péter u. 4, 7634 Pécs, Hungary
2   Dula Grape & Wine Advisory Ltd., Eszterházy Tér 9, 3300 Eger, Hungary
3   Institute of Viticulture and Oenology, Hungarian University of Agriculture and Life Sciences, Villányi út 29-43, 1118 Budapest, Hungary
*   Correspondence: deak.tamas@uni-mate.hu

**Abstract:** The aim of this review is to provide readers with an integrated knowledge on black rot of grapes, based on a critical survey of previous and recent studies of scientific importance. The current state of the art and perspectives of science are presented, not only on the genetic determinants of grapevine resistance to black rot, predictive models of black rot epidemics, but also on the potential of metabolomics to explore black rot-grape interactions and shorten plant breeding processes. Numerous complications of disease management and ambiguities in phenotype-classification are highlighted, and by exploring the limitations and inconsistencies of previous studies, insights into key dilemmas and controversial findings are also provided, suggesting future research directions. Much research has been conducted, but biochemical and molecular studies of the true interactions between grapevine and *Guignardia bidwellii* are still rarity.

**Keywords:** disease control; resistance; host-pathogen interaction





## 1. Introduction

Grapes have been cultivated for thousands of years and are of great economic and cultural importance. According to the International Organization of Vine and Wine, 85 million tons of grapes were produced worldwide in 2019, and the global vineyard surface area is estimated at around 7.3 million hectares [1].

Vitaceae includes 15 genera and about 900 species [2], but only one of them, *Vitis vinifera*, which takes account of almost all varieties, is cultivated for high quality fruits, and is mainly used for wine production. Compared to wild *Vitis* species, however, these varieties are more affected by environmental factors such as biotic (damage caused by fungi, insects, nematodes, bacteria, etc.) and abiotic (e.g., temperature, heavy metals, salinity, etc.) stresses, which can result in significantly lower and limited yields. These environmental factors have threatened yields since the beginning of agricultural and horticultural production, the emergence of new and almost forgotten grapevine pathogens due to climate change, as well as societal expectations and stricter regulation of pesticide use, are creating a new set of challenges for the European grapevine and wine sector. A good example is that since the turn of the millennium, vineyards of Central European countries have been struggling with the black rot of grapes (caused by *Guignardia bidwellii*), a disease known for a long time in scientific literature but relatively new in practice.

*G. bidwellii* was identified in North America in 1853 and spread to Europe in the late 19th century following powdery mildew (caused by *Erysiphe necator*) and downy mildew (caused by *Plasmopara viticola*). European grapevine cultivars are susceptible to black rot [3], and without adequate crop protection and under warm and humid conditions, emergence of black rot can be expected. In heavily infected vineyards, sometimes up to 100% yield loss can occur [4].

Viticulture is always facing new challenges to which producers, manufacturers and researchers must find answers. Black rot is one of those problematic threats that is reemerging today. The recent knowledge on black rot of grapevine, the main current problems, and proposed solutions in the field of plant protection, together with the results of plant breeding were summarized. Special attention was paid to the biochemical characterization of the interactions between grapevine and black rot.

## 2. Classification, Nomenclature, Emergence and Distribution of Black Rot

Black rot of grapes is caused by *Guignardia bidwellii* (Ellis) Viala and Ravaz [5] (asexual stage: *Phyllosticta ampelicida* [Engleman] Van der Aa) [6]. The fungus belongs to the phylum Ascomycota, class Dothideomycetes, order Botryosphaeriales, family Botryosphaericaceae and genus *Guignardia* [7]. Conidial states of *Guignardia* belong to the genus *Phyllosticta*, which is an important genus of various plant pathogens [8], but in the scientific literature the pathogen responsible for black rot of grapes is mostly referred to as *G. bidwellii*.

To date, three forms of *G. bidwellii* with different host affinities have been described. *G. bidwellii forma specialis euvitis* is pathogenic to the American bunch grape species of the section *Vitis*, and to *V. vinifera*. *G. bidwellii* f. sp. *muscadinii* is pathogenic to *V. rotundifolia* and *V. vinifera* and *G. bidwellii* f. sp. *parthenocissi* is pathogenic only to species of *Parthenocissus* (Vitaceae). *G. bidwellii muscadinii* also differs from *G. bidwellii euvitis* morphologically [9].

Black rot of grapevine is native to North America. Significant damage was first recorded in Ohio in 1848 [10]. In Europe, it was first detected in 1885 in the south of France [5], and then spread to the wine-producing regions of neighbouring Germany and Italy, and later to the former Yugoslavia [11–13]. However, it was only towards the end of the 20th century that major outbreaks were reported from Germany and Switzerland [14,15], and increasing outbreaks were also observed in Italy, Portugal, Austria, Luxembourg, Romania and Hungary [4,16–19]. In addition to North America and Europe, it has also been described in Central and South America, Asia, Africa, Oceania, and its importance has been recognized by the Australian authorities, making grape black rot a disease that has an almost global distribution [20]. Grape black rot was probably introduced to other wine-producing regions of the world by infected propagating material [21].

## 3. Genetic Diversity and Population Biology of Black Rot Agent

According to Zhang et al. [22] and Zhou et al. [23] grape black rot is considered to be a species complex, of which at least four can be associated with the disease: *P. ampelicida*, *P. parthenocissi*, *P. partricuspidatae*, *P. vitis-rotundifolia* [22,23]. However, *P. ampelicida* is still the most important species present in cultivated grapes, both for *V. vinifera* cultivars and interspecific hybrids [24].

Different genotypes of the black rot agent can be identified based on nuclear genetic markers like ITS, beta-tubulin and calmodulin gene sequences [25]. This specificity allowed the generation of ITS based *P. ampelicida* diagnostic markers [26]. The differentiation of *P. ampelicida* populations between North America and Europe, and within Europe between various vineyards or even within blocks of vineyards can be detected with SSR markers [25,27,28]. Based on the population structure derived from SSR markers, the already formulated hypothesis of several distinct introduction events of grapevine black rot to the continent is supported [25]. Recently, a high quality draft genome of *P. ampelicida* has been published [29].

## 4. Disease Cycle and Environmental Conditions for Development

Black rot of grapes is a typical polycyclic disease that can cause polyetic epidemics. It is capable for several infection cycles in a season and the inoculum produced in one season can be carried over to the next one, leading to the accumulation of inoculum over the years.

*G. bidwellii* is a hemibiotrophic [30–32] endoparasite that undergoes an asexual and sexual life cycle. Spermagonia with bacilliform spermatia and ascogonial stromata are produced towards the end of the growing season on mummified berries and in cane lesions,

forming the overwintering sexual fruiting bodies of *G. bidwellii*, the pseudothecia. However, the fungus overwinters not only in the form of pseudothecia, but also in asexually produced pycnidia on many plant parts, especially on fruit mummies, buds, cane bases, on the vine, or on infected plant parts lying on the ground.

Disease cycle of black rot is shown with symptoms schematically in Figure 1. In spring, pycnidia produce conidia and pseudothecia produce ascospores that, under favourable conditions, infect all young green tissues (leaves, shoots and bunches) of a susceptible vine [9,33,34]. The vast majority of ascospore discharge have been recorded by the end of the bloom period, though drought during the prebloom and bloom period can delay ascospore depletion until rains resume. Moreover, ascospores from mummies left on the ground can be discharged not just in the beginning of the growing season but from bud break through mid-summer and they continue to be produced into late summer from mummies that are retained in the canopy, thus providing continuous primary infection. Since the ascospores are actively ejected and airborne, they allow black rot to spread over long distances. They can infect susceptible hosts from 100 m or even a longer distance from the nearest inoculum source (e.g., wild grapevines); however, their density is diluted with distance [9,14,35].

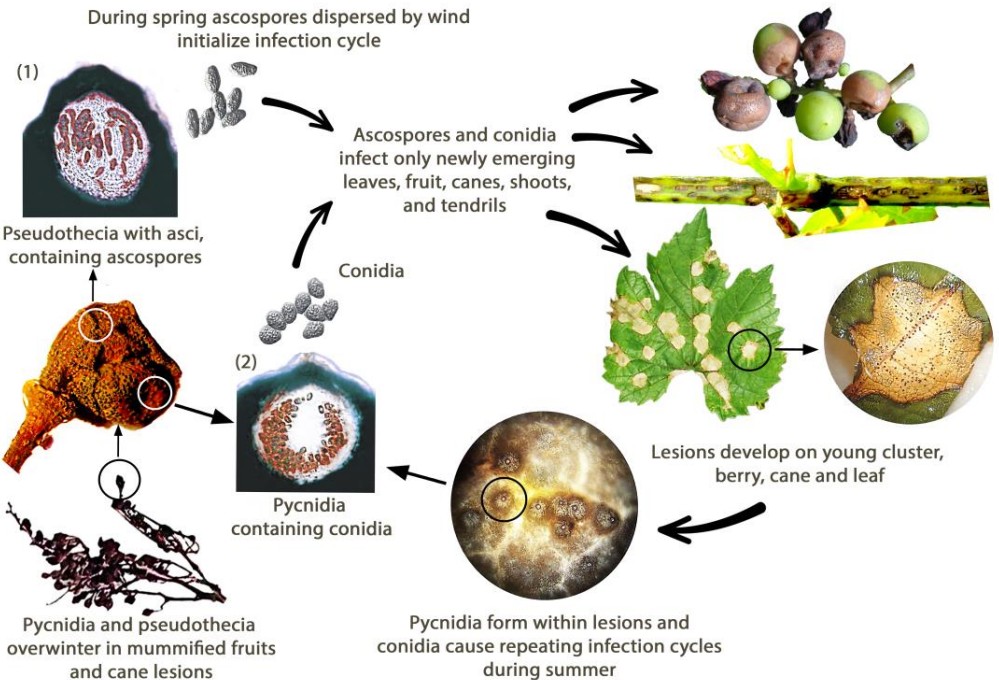

**Figure 1.** Disease cycle of black rot of grapes (1) (2) [6].

Unlike ascospores, conidia are released passively in a white, slimy cirrus from which they can be splashed away in large quantities by rain only over short distances. They serve as secondary inoculum and are responsible for rapid and repeated spread of the disease during the growing season. Ascospores and conidia are both sensitive to desiccation; however, pycnidia can produce conidia even after three months of low humidity [36]. Another interesting difference is that as little as 3 mm of rain is sufficient to discharge ascospores, while conidia release requires at least 10 mm of rain [9,37].

A constantly moist leaf surface is required for the infection process to begin [37,38], and the lower the temperature, the longer this process takes (Figure 2). When these conditions are met, ascospores and conidia can germinate in 36–48 h and then penetrate young plant cells. About 14 days later, at the sites of the penetration brownish lesions are observed on leaves, shoots and berries on which black asexual pycnidia appear [39,40]. These pycnidia produce again conidia, which can repeatedly attack all parts of the plant during the growing season. Infections begin to decline in late July and disappear by late August [41]. Fully

developed adult leaves and berries after the onset of berry ripening are not susceptible to infection, indicating age resistance [9,35]. The fungus can persist on infected plant debris for up to 2 years [42].

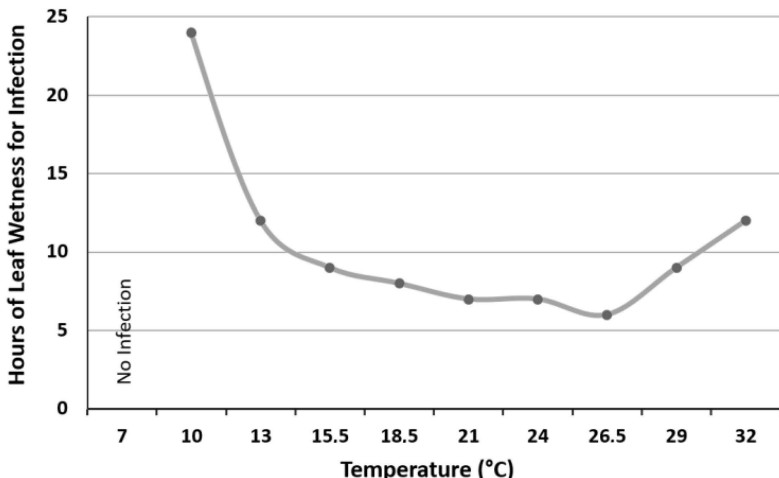

**Figure 2.** Duration of continuous leaf wetness necessary for infection by the black rot fungus at different temperatures [43].

## 5. Symptomatology

Following infection of the grapevine plant, initiated by one or two hyphae originating from the base of the appressorium penetrating the cuticle, hyphae grow mainly between the cuticle and the anticlinal cell walls, forming a dense, two-dimensional mycelium [44], all without any external signs on the host plant. The fungus spends at least ten to eleven days in this phase [45]. Black rot is particularly characterized by this two-dimensional hyphal growth, which is typical of its long biotrophic phase. Only after the transition to necrotrophic stage does the mycelium colonize additional cell layers. Another characteristic of black rot is that it does not form haustoria, and does not directly invade or destroy neighbouring cells [46]. Approximately 12–14 days after infection, the hyphal nets become denser and start to develop into pycnidia, while the surrounding tissue becomes necrotic [44].

Black rot shows symptoms on all the growing green parts of the vine (Figure 1) [47]. 1–2 weeks after infection, small, round or slightly segmented, brownish-reddish spots with dark edges become visible on the leaves. The black pycnidia (59–196 μm in diameter) are sometimes arranged in concentric circles and appear at the edges of the leaf spots and if the susceptibility of the host plant is high, hundreds or even thousands of pycnidia may be found on the spots (Figure 3A). Several spots may form on one leaf, sometimes fused together (Figure 3B).

Lesions can also develop on the petiole, rachis (Figure 3D), peduncle, pedicels, shoots (Figure 3C), and tendrils. On the petioles and pedicels, they appear as small, darkened depressions, which soon turn black. Occasionally, lesions girdle the petiole and kill the entire leaf. Shoot infections appear as larger, darker, oval and slightly sunken elongated black cankers ranging in length from 1 mm to 20 mm [9]. The shoots have spots, with pycnidia on the surface of the torn skin tissue (Figure 3C). Shoot and petiole lesions often contain abundant pycnidia.

The tendrils and all green parts of the cluster can also be infected [48]. Generally, in sprayed vineyards in Central Europe, black rot symptoms on bunches may occur shortly after cap-fall, in critical years black rot can infect even earlier (Figure 3D), but usually only in untreated vineyards.

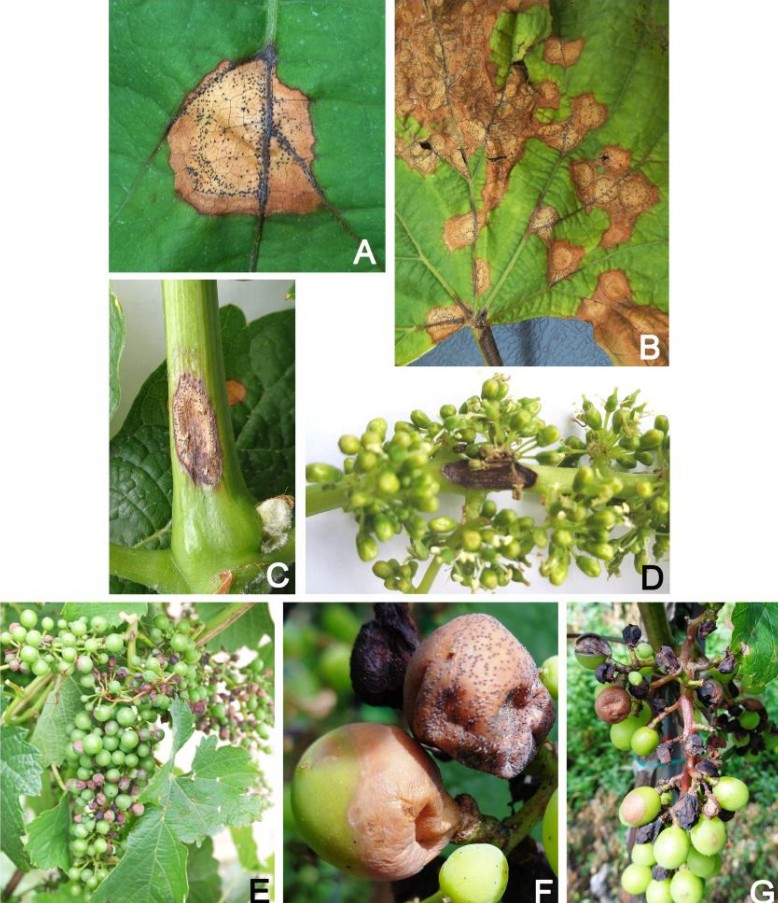

**Figure 3.** (**A**) Leaf spot with abundant black pycnidia. (**B**) Leaf spots fused together. (**C**) A lesion on torn skin tissue of a shoot. (**D**) Rachis infection. (**E**) Partial cluster infection. (**F**) Berries turning brown and mass of pycnidia become apparent on berries. (**G**) Alternating healthy and discoloured diseased berries.

The importance of the pathogen is increased by the fact that it can cause complete cluster death, although the cluster is rarely completely infected (Figure 3E). The most affected are the green berries, which initially show faded spots that turn brown within 24–48 h, their surface becomes wrinkled, and within 1–2 weeks their surface feels rough due to the mass of pycnidia (Figure 3F). Eventually the diseased berries dry out and darken. It is also characteristic that black rot does not attack all berries in the bunch at the same time, so that diseased berries alternate irregularly with healthy berries (Figure 3G). Mummified berries are usually firmly attached to the bunch [9]. Inside black rot-infected berries, pycnidia are never found on the seed coat.

Depending on weather conditions, vines are most susceptible to black rot 1–5 weeks after flowering [49]. Although *G. bidwellii* shows well-defined symptoms on vines, these can sometimes be confusing to the observer. For instance, until the appearance of propagules, the symptoms of black rot are similar to those caused by herbicide, fertilizer scorch or other fungi that cause wilting and rotting [50]. Sunburn can also lead to similar symptoms. Furthermore, typical grapevine diseases infecting at the same time and showing similar symptoms on *V. vinifera* may also show a variable symptom complex between climates and continents. For example, in central Europe, black rot on bunches and berries can show similar symptoms to those of grape white rot (caused by *Coniella diplodiella*), grape grey mould (caused by *Botrytis cinerea*) and grape downy mildew (caused by *Plasmopara viticola*) [40]. In Australia, similar symptoms with anthracnose (caused by *Elsinoë ampelina*) [51], Phomopsis cane leaf spot and fruit rot (caused by *Phomopsis viticola*) are highlighted [52]. In the United

States, the early-season diseases are black rot, Phomopsis fruit rot, and bitter rot (caused by *Greeneria uvicola*) [53].

## 6. Host Plant Community and Black Rot Resistance

### 6.1. Host Plants

Host plants include several species of the genera *Vitis*, *Parthenocissus*, *Cissus* and *Ampelopsis* [54,55] but among them, cultivars of *Vitis vinifera* are the most susceptible. Native American species such as fox grape (*Vitis labrusca*), muscadine (*Vitis rotundifolia*) and Arizona grape (*Vitis arizonica*) are also susceptible. Other important host plants are the Asian Amur grape (*Vitis amurensis*), California wild grape (*Vitis californica*), the American five-leaved ivy (*Parthenocissus quinquefolia*) and the Asian Boston ivy (*Parthenocissus tricuspidata*) [56].

Fortunately, there are nearly resistant *Vitis* species such as riverbank grape (*Vitis riparia*); sand grape (*Vitis rupestris*) which was also used for breeding several French-American hybrids; mustang grape (*Vitis mustangensis*, syn.: *Vitis candicans*) and frost grape (*Vitis vulpina*, syn.: *Vitis cordifolia*) [52].

According to Wilcox [57], all *V. vinifera* cultivars are considered sensitive to black rot infection, but there may be variation in susceptibility between cultivars. There are indeed differences in susceptibility to black rot among members of the Vitaceae family, and species can be classified according to it. Dissimilarities in susceptibility among grape cultivars may be due to different levels of genetically determined physiological resistance, but differences in the duration of susceptible phenological stages [39] and the number of days to symptom onset are also cultivar-specific [58].

### 6.2. Genetic Resources of Black Rot Resistance in Grapevine

Grapevine breeding against fungal diseases especially downy mildew and powdery mildew dates back more than 150 years [59] but unfortunately, many of the commercially available mildew resistant cultivars are susceptible to black rot, although the severity of the symptoms might vary (Table 1). For example, the mildew resistant American hybrid 'Isabella' of *V. labrusca* origin can show severe black rot symptoms [60], most probably because *V. labrusca* does not show considerable resistance against black rot [61]. *V. rotundifolia* (syn. *Muscadinia rotundifolia*) is an important resistance source in modern grapevine breeding because it has well defined immunity level resistance against powdery and downy mildew. However, *G. bidwellii* is able to infect muscadine grapes as well [62].

The most widely studied genetic resource of resistance to black rot in grapes is *V. riparia*. Several mapping populations of *V. riparia* and *V. cinerea* have been used to search for resistance to black rot [3,63,64]. Although *V. cinerea* was originally thought to be the source of black rot resistance [64], more recently it has been accepted that the resistance is derived from *V. riparia* [65].

Based on the susceptibility levels of different interspecific crosses, the widely used Seyve-Villard 12375 hybrid (syn. 'Villard blanc') seems to lack an effective resistance source against black rot, while Seibel 4986 (syn. 'Rayon d'or') and in particular Seibel 4643 (syn. 'Roi des noirs') can pass on high disease resistance to offspring [58]. As such, Seibel 4986 and its descendant 'Csillám' ('Kékfrankos' × 'Rayon d'or') are promising genetic resources in grape breeding for black rot resistance [66]. 'Csillám' shows no leaf symptoms and only mild symptoms on the berry, where the lesions affect only the berry skin, the flesh remains unaffected [67].

In addition to North American species, the European wild grape, *Vitis vinifera* subsp. *sylvestris* (Willd). Hegi [68] (hereinafter referred to as *V. sylvestris*) also appears to be a promising source for breeders. Despite the fact that Eurasian *Vitis* species were not selected under the pressure of fungal diseases native to North America, resistance is documented against powdery mildew in European and Asian grapevine species [69–71]. *V. sylvestris* seems to harbour resistance against black rot as well, which could be introduced in cultivated varieties. Indeed, Schröder et al. [72] identified a *V. sylvestris* genotype that

exhibited a very high resistance level to black rot, making it a good candidate source of resistance for grape breeders.

Several studies have been published evaluating the susceptibility of grapevine cultivars to black rot (Table 1). Although there are no *V. vinifera* cultivars with high levels of resistance, interspecific hybrids may show a wide range of resistance levels to black rot. However, many of the powdery and downy mildew-resistant varieties among the hybrids are susceptible to black rot.

**Table 1.** Resistance and susceptibility levels of different grapevine cultivars to black rot.

| Cultivar | Hausmann et al. [1] [3] | Loskill et al. [2] [73] | Rex [1] [74] | Roznik et al. [3] [58] | Tomoiaga and Chedea [4] [75] |
|---|---|---|---|---|---|
| Amurg | | | | | T |
| Astra | | | | | S |
| Baron | | MS | | | |
| Beta | 8 | | | | |
| Bianca | 5 | | | HS | |
| Blasius | | | | | T |
| Börner * | | | 9 | | |
| Bronner | | MS | | | |
| Brumariu | | | | | T |
| Cabernet carbon | | MS | | | |
| Cabernet carol | | LS | | | |
| Cabernet cortis | | MS | | | |
| Cabernet sauvignon | | HS | | | |
| Campbell early | 5 | | | | |
| Carman | 9 | | | | |
| Catawba | 7 | | | | |
| Cayuga white | 6 | | | | |
| Champanel | 8 | | | | |
| Chancellor | 9 | | | | |
| Chardonnay | | | | HS | |
| Clinton | 9 | | | | |
| Cloeta | 9 | | | | |
| Concord | 6 | | | | |
| Cynthiana | 6 | | | | |
| Csillám | | | | SR | |
| De Chaunac | 9 | | | | |
| Delaware | 5 | | | | |
| Emerald | 5 | | | | |
| Esther | | | | HS | |
| Etta | 8 | | | | |
| Felicia | | | 7 | MR | |
| Feteasca alba | | | | | S |
| Feteasca regale | | | | | S |
| Fredonia | 9 | | | | |
| Furmint | | | | HS | |
| Hanover | 9 | | | | |
| Helios | | MS | | | |
| Iordana | | | | | MT |
| Ironclad | 9 | | | | |
| Isabella | 4 | | | | |
| Isaura | | | | HS | |
| Jefferson | 3 | | | | |
| Johanniter | | HS | | | |
| Lemberger | | | 1 | | |
| Malverina | | | | MR | |
| Manito | 5 | | | | |

| Cultivar | Hausmann et al. [1] [3] | Loskill et al. [2] [73] | Rex [1] [74] | Roznik et al. [3] [58] | Tomoiaga and Chedea [4] [75] |
|---|---|---|---|---|---|
| Mars | 6 | | | | |
| Merlot | | HS | | | |
| Merzling | | LS | 9 | MR | |
| Mills | 5 | | | | |
| Missouri riesling | 9 | | | | |
| Moldova | | | | MS | |
| Monarch | | MS | | | |
| Muscat Ottonel | | | | | S |
| Müller-Thurgau | 1 | HS | 1 | | |
| Nero | | | | MS | |
| Neuburger | | | | | MT |
| Pearls | 5 | | | | |
| Pinot gris | | | | | S |
| Pinot noir | 1 | | | | |
| Pinotin | | HS | | | |
| Primitivo | | | 3 | | |
| Prior | | MS | | | |
| Radames | | | | | T |
| Reberger | | | 3 | | |
| Regent | | HS | 3 | | |
| Riesling | 1 | HS | | | MT |
| Rommel | 8 | | | | |
| Rubin | | | | | T |
| Selena | | | | | MT |
| Sauvignon blanc | | | | | S |
| Seyval blanc | | | | HR | |
| Solaris | | LS | | | |
| Spätburgunder | | HS | | | |
| Suelter | 9 | | | | |
| Suzy | | | | MS | |
| Teréz | | | | HR | |
| Traminer Rot | | | | | MT |
| Triumph | 6 | | | | |
| Trollinger | | | 1 | | |
| Viktória gyöngye | | | | MS | |
| Villard blanc | | | 7 | | |
| Villaris | | | 5 | MR | |
| Wapanuka | 7 | | | | |
| Welschriesling | | | | | S |
| Xlnta | 9 | | | | |
| Zalán | | | | HS | |

Please note that phenotyping of black rot resistance is challenging because of the complex disease cycle and strong dependence of symptom development on plant developmental stages [9], resulting in diverse phenotyping scales in different papers. To avoid any misleading classification, the original resistance categories published by the authors are reported. [1] 1–9: resistance categories according to Hausmann et al. [3], 9 being the highest level of resistance. [2] HS—high susceptibility, MS—medium susceptibility, LS—low susceptibility. [3] HS—high-level susceptibility, MS—moderate susceptibility, MR—moderate resistance, HR—high-level resistance, SR—symptomless resistance. [4] T—tolerant, MT—middle tolerant, S—susceptible. * Rootstock cultivar.

Additionally, to the cultivars listed in Table 1, there are studies comparing grapevine black rot susceptibility/resistance [76,77], but these are comparing susceptibility level of a selected pool of cultivars and they are not assigning actual resistance categories to the genotypes.

*6.3. Genetic Factors Determining Resistance to Black Rot in Grapes*

The hemibiotrophic nature of *G. bidwellii* and the slow and complex disease progression make phenotyping of resistance levels difficult, which in turn makes reliable genetic mapping challenging. The first mapping study on resistance of grapevine to black rot was published by Dalbó et al. [63], using a mapping population derived from a cross between 'Horizon' (a complex interspecific hybrid) and Illinois 547-1 (*V. rupestris* × *V. cinerea*). The authors concluded that two smaller QTLs of powdery mildew resistance on linkage groups 16 and 18 overlap with black rot resistance QTLs. One of these QTLs (on LG 16) also overlapped with a QTL for resveratrol degradation in the susceptible parent [63] indicating the importance of phytoalexins in the host-pathogen relations.

The next attempt to identify the genomic regions responsible for resistance to black rot was carried out on a mapping population derived from a cross between V3125 ('Schiava grossa' × 'Riesling') and 'Börner' [64], which, like Illionis 547-1 [63], is a hybrid of *V. cinerea*. Rex et al. [64] identified two major QTLs on linkage groups 14 (*Rgb1*) and 16 (*Rgb2*). The presence of *Rgb1* was also confirmed by fine mapping of the region and the authors identified two markers closely linked to *Rgb1*: Gf14-41 and Gf14-42.

The mapping population of GF.GA-47-42 × 'Villard blanc' revealed again an important QTL in the same region of linkage group 14 [3]. As such, the QTL from LG14 of *V. riparia* is considered an important candidate locus.

Previously, the resistance of 'Börner' (*V. riparia* Gm183 × *V. cinerea*) to black rot was thought to be derived from *V. cinerea*, although *V. riparia* has not been ruled out as a possible source of resistance [64]. However, recent characterization of two known QTL (*Rgb1* and *Rgb2*) conferring resistance to black rot in 'Börner' clearly identified *Vitis riparia* as the origin of both resistance loci [65].

Interestingly, Bettinelli et al. [78] mapped the main QTL of black rot resistance in a Merzling × *V. vinifera* hybrid family on LG14 as well, although Merzling inherited its black rot resistance most probably from *V. rupestris*.

Further work is needed to identify other sources of genes that confer highly effective resistance, which can be applied in gene pyramiding to ensure long lasting defence against grape black rot. The genetic constitution of resistance in *V. sylvestris* [72] and American black rot resistant *Vitis* species still needs to be investigated to identify new QTLs and to further develop the tools available for marker-assisted selection.

## 7. The Control of Black Rot

*7.1. Changes in Cultivation Practices in Central Europe*

*G. bidwellii* has been spreading worldwide since the beginning of the 20th century. It is an increasingly important pathogen, especially in Central European vineyards. The increasing prevalence and growing economic importance of the pathogen in lowland areas, particularly in organic farming are mainly due to changes in cultivation practices that facilitate the accumulation of the pathogen's infectious inoculum [79]. The main reasons can be summarized as follows:

- With the reduction of manual labour and increasing mechanization (e.g., mechanical pruning and mechanical harvesting), infected grapes and grape parts remain frequently on the vine.
- The decline of cover crops and inter-row cultivation has led to the spread of mechanical shredding of prunings. Thus, infected plant parts remain between vine rows, on the soil or on the surface of the inter-row vegetation, facilitating the persistence and accumulation of the infective material within the plantation.
- Dense vine spacing and low vine heights create more favourable microclimatic conditions for pathogens, resulting in an increased risk of plant diseases.
- Organic farming is spreading, but with no effective control options against black rot, as the permitted copper and sulphur-based products are ineffective. Systemic pesticides are effective against black rot but banned in organic farming. Therefore, the control of black rot in organic crops is slowly becoming unmanageable.

- Strong increase in the cultivation area of inter- and intraspecific varieties resistant to powdery mildew and downy mildew, but highly susceptible to black rot, in lowland viticultural regions and in organic farming, creating an ecological niche for emerging pathogens such as *G. bidwellii*.
- The official withdrawal of certain active substances e.g., sterol biosynthesis inhibitors such as demethylation inhibitors, strobilurin, and contact dithiocarbamates, has made it increasingly difficult to control the pathogen effectively in Integrated Production and in European national agricultural programs such as Agri-Environment Scheme (AES) that supports biodiversity, quality of water, air and soil.
- The growth of abandoned and uncultivated vineyards is a major and growing problem in Central Europe, as black rot and other diseases spread unhindered in these areas. Infectious material can easily be carried by wind from these areas to areas still free of infection.

The above-mentioned changes in cultivation techniques can pose a real threat to vineyards, not only because of black rot, but also because they can facilitate the spread of other common grapevine diseases and even the emergence of quarantine diseases such as "Pierce's disease" and "flavescence dorée".

To achieve successful black rot control, strategies should combine sanitary measures, cultural techniques, growing cultivars with reduced susceptibility and the use of effective fungicides [15,57]. The aim of the treatments is to prevent the accumulation of infective material by all possible means, because it is not the mass of infective material accumulated in a given year, but the mass of infective material accumulated over several years that pose serious risk of epidemics.

### 7.2. Prevention of Black Rot Infection by Agrotechnology

- After harvesting, the amount of infective material can be reduced below critical level by cutting and burning mummified clusters [80] and by turning infected plant debris that falls to the ground during pruning into the soil [57,81]. Experiments have shown that this method is very effective when used in heavily infected areas for several, but at least for two consecutive years [42].
- The removal and destruction (e.g., burying) of the first infected, symptomatic leaves found during shoot thinning is a beneficial method for cluster protection.
- An airy, thin canopy and keeping rows free of weeds will ensure that foliage dries quickly after rainfall, reducing the risk of infection. Low cover cropping can also reduce relative humidity in the plantation.
- Nearby abandoned vineyards should be treated or eradicated to reduce the chance of infection.
- Any measure that limits or prevents physical damage to grapes and clusters is important.
- A balanced supply of nutrients reduces the susceptibility of the vine to diseases. It is strongly advisable to avoid excessive nitrogen supply to prevent extreme growth of vegetative parts.

In order to preserve the integrity of nature and meet the needs of humanity in a sustainable way, it is important to reduce and eliminate risk factors that can lead to the development of diseases in cultivated plants, especially in vineyards where the use of pesticides is very high. Indeed, even chemical treatments that are often undesirable for nature and the environment can only be effective if preventive agrotechnical procedures are applied with maximum attention.

### 7.3. Biological Control of Black Rot

In organic farming, the list of products that can be used for control is limited, so great attention must be paid to preventing infection and limiting pathogenic spores through appropriate agrotechnical and phytotechnical measures or by cultivating resistant varieties.

Based on French and American experiments, Linhart and Mezey [82] emphasized in one of the first studies on the control of black rot of grapes, the 80–90% effectiveness of the

Bordeux mixture in control, stressing the importance of the proper timing of treatments. Unfortunately; however, the very frequent application of sulphur and copper preparations in organic crops is nowadays no longer effective in controlling black rot [15,73]. In addition, the use of copper in European organic farming is declining significantly [83] due to its high accumulation in soil [84]. The issue of authorization of certain products and substances for use in organic production are covered in 'Commission Implementing Regulation (EU) 2021/1165' [85].

A number of research groups are investigating alternatives to chemical crop protection in organic farming, using various microorganisms or their metabolites, other plant extracts or other natural substances that may have fungicidal activity. Although, a disadvantage of using organic fungicides is that they play a protective rather than curative role. Thus, the appropriate timing of application for organic products used to manage plant disease is critical [86]. Rutto et al. [87] came to the same conclusion with the addition that their findings highlight the importance of breeding for disease resistance. They also confirmed that proper cultivar choice might be one of the most important factors in organic viticulture.

Versatile and interesting groups of antimicrobial agents are natural products containing saponins, since the presence of saponins in plants has been shown in several cases to be associated with resistance to plant pathogens [88]. First, the activity of *Yucca schidigera* extract against *G. bidwellii* was demonstrated in potted grapevines [89], followed by Molitor et al. [90] who reported the efficacy of soap berry extract on grapevines. However, Travis et al. [89] found that yucca extract, which worked well in greenhouse screening trials, was ineffective in the field at suppressing black rot of berries under high and moderate pressures when applied to whole vines at 10-day intervals.

Koch et al. [91] demonstrated the efficacy of root extracts of *Hedera helix* and primula against grapevine black rot. The primula extract showed a stronger level of inhibition, completely inhibiting the germination of conidia on grape leaves, similar to the Polyram® WG agent (metiram). Under glasshouse conditions, the efficacy of alcoholic root extracts of both plants at concentrations of 1 and 0.5% has been demonstrated to be above 90%.

Stafne et al. [92] observed differences between two organic spray treatment suggesting that product formulation may affect efficacy. They found that a suspension concentrate of a combination of *Bacillus subtilis* and basic copper sulphate had a significantly stronger effect on black rot than the same combination in the form of a wettable powder.

Pálfi et al. [93] demonstrated that the culture filtrate of *Aspergillus niger* could be used for the control of some important fungal pathogens of grapevine. The agent exhibited high antifungal activity against *G. bidwellii*; however, the filtrate was not able to penetrate leaf cuticle; therefore, cannot be applied as curative fungicide against subcuticular mycelia. Nevertheless, its high efficiency against epicuticular mycelia, fruiting bodies and spores can be useful in the prevention of black rot infections.

### 7.4. Models for Predicting Black Rot Epidemics

Plant disease epidemiology is the cornerstone of plant protection systems. It shapes decisions and evidence-based practice by identifying risk factors for plant disease and targets for preventive and curative measurements. As applied epidemiology, disease forecasting is the prediction of the time of a probable outbreak. Effective pest control requires system approach, which involves well-organized teamwork, precision, speed, and non-destructive practices; therefore, it is crucial to develop applicable forecasting system to allow the selection and timing of the most appropriate control measures. Accurate forecasting of any fungal attack in vineyards can greatly reduce the amount of fungicide applied, but many factors need to be known in advance and all of them need to be taken into account to make accurate predictions. It requires frequent field observation, collection of data about pathogen, weather, and crop varieties and their correlations. With the computing, information- and remote sensing technology background of the 21st century, one would assume that these factors could be easily used as input data to calculate the time of a plant-disease outbreak. However, after a thorough review of big data analytics

techniques, Fenu and Malloci [94] pointed out that use of disease prediction algorithms is still in its infancy and there are still many hurdles to overcome. Their study shows that today the most studied forecast models fall into three categories: the first category based on weather data, the second based on image processing and these are the most explored; the third category is based on distinct types of data coming from various heterogeneous sources. Studies of the first and second categories indicated that the exclusive use of a single data source is not sufficient to build models capable of capturing and predicting the variability of a disease in the field. To increase the stability and generalization capabilities of the algorithms, the integration of multiple data sources, as well as the inclusion of more information such as previous infections, susceptibility, relative humidity levels, leaf wetness, plant age, cultivar, growth phase and soil characteristics has been suggested.

In general, black rot of grapevine is not a problem in chemically treated plantations because frequent fungicide treatments against other more common diseases such as powdery mildew, downy mildew, botrytis, etc. prevent its damage, but in organic farms black rot can cause serious damage, so a reliable forecasting system is essential.

First, Spotts [43] reported a simple model based on leaf wetness duration-temperature combinations (Figure 2) and the effect of variable and constant temperature on the severity of grapevine foliar infection caused by *G. bidwellii*. This was not a mathematical model but proved to be reliable enough for determining infection periods. Using a similar approach, Maurin et al. [95] developed a prediction model using relative humidity, precipitation, and temperature data to predict the risk of black rot infection. The model was able to predict favourable periods for the pathogen in two test years.

The first computer-based program to forecast grape black rot infection was developed by Ellis et al. [96]. They used a Reuter-Stokes Disease Predictor RSS-411 microprocessor, which was programmed to predict black rot infection periods from the parameters established also by Spotts [43]. The platform was effective in determining infection periods and was used to time curative fungicide applications.

Northover et al. [97] developed a model in the United States based on daily average temperature, calculated from the minimum and maximum temperature, with a temperature threshold of 7 °C and 26 °C. The time required from infection to formation of new inoculum was defined in degree-days (DD) at different stages of the growing season. It was found that 196–248 DD were required before symptoms on the leaf appeared, and that spore exudation from leaf spots peaked at 400 DD.

Smith [98] has developed an advisory system based on the susceptibility to black rot at different stages of vine growth.

On average, the application of this system reduced the number of fungicide treatments by about 45%. Smith and Sutherland [99] improved the system by developing a second-order equations that calculated the number of hours of humidity (relative humidity > 85%) required for black rot infection as a function of daily temperature. The resulting mathematical model has become part of the "Mesonet Grape Black Rot Advisor" system in the United States [100]. For detailed prediction of black rot epidemics, Rossi et al. [101] developed a new mechanistic, dynamic, weather- and phenology-driven model. The model accounts for the complexity of the *G. bidwellii* life cycle, specifically the maturation of ascospores and conidia in overwintering fruiting bodies, spore release and survival, infection frequency and severity, incubation and latency periods, lesion development, pycnidia production, and infection duration. After a thorough validation [102], the authors showed that this model is very accurate and robust in predicting infection periods and black rot epidemic dynamics and can be used for scheduling fungicide sprays in vineyards.

To date, only a few successful grape black rot decision support applications have been developed that integrate multiple data sources, and one of them is VitiMeteo Black Rot [48]. VitiMeteo Black Rot was developed based on existing sub-models and integrated with the existing VitiMeteo forecasting and decision support platform. It simulates the relative susceptibility of grape clusters, the occurrence and severity of infection events and the duration of the incubation period based on local weather data and 5-day weather forecasts.

VitiMeteo Black Rot is freely available for several locations in Germany, Luxembourg and Austria on the internet [103]. In Hungary, BASF's computer-aided forecasting system [104] also takes into account several variables and can provide useful information for predicting black rot infection.

*7.5. Chemical Crop Protection*

Control of black rot relies mainly on the use of fungicides. Unlike other diseases, chemicals can effectively control black rot after infection due to the long incubation period of *G. bidwellii*. In theory, fungicidal treatments are unnecessary before full anthesis, as the shedding of the calyptras (flower caps) removes all previously applied active ingredients (even absorbed sterol inhibitors) from the bunches, leaving the freshly bound berries completely unprotected. In line with this, Hoffman and Wilcox [49] suggest that control should be applied during the most susceptible period of the cluster, from full bloom until 6-7 (in extreme cases 10) weeks after flowering. Somewhat to the contrary, according to Wilcox [57] excellent control can be obtained when fungicides are applied before full anthesis, i.e., from immediate pre-bloom stage through 4 weeks post bloom and in fact, spraying should start at least 2 weeks before flowering if the disease was severe the previous year. Moreover, Hoffman et al. [105] found that, applications of myclobutanil immediately prior to bloom plus 2 and 4 weeks later provided virtually complete control of fruit rot while reducing the standard number of seasonal fungicide applications by 50% or more. However, despite its good efficacy, the active substance myclobutanil has been officially withdrawn in Europe. Fungicide treatments should be repeated depending on the temperature and the distribution of precipitation. In humid conditions, rotations of 12–14 days are recommended [17]. The priority during spraying is to protect the clusters. The use of fungicides against black rot is not necessary if the berries have already accumulated more than 5% sugar [21].

*G. bidwellii* attacks all young green tissue, so green berries of susceptible varieties are at high risk of infection, thus the control of black rot might be improved by focusing on the variable effect of fruit age on the incubation period length of black rot. Indeed, Hoffman et al. [39] found that age-related berry resistance was manifested as both a decline in susceptibility and a significant increase in incubation period length. Their results suggest that susceptibility and the associated need for fungicide control ends much earlier, 10 weeks after flowering.

The assessment of the need for protection is further complicated by the fact that the number of days after infection for symptom appearance are also cultivar-specific [58].

Several important groups of well-known and widely used active ingredients can be used to control downy mildew and/or powdery mildew on grapevines [15], which are well-established conventional products and have so far kept black rot damage below detection limit, too. In addition to epidemiological factors, it is very important to understand the specifics of these when trying to devise efficient disease management strategies against black rot. There are three main groups of active substances that are commonly used against powdery mildew and/or downy mildew, and can also potentially be used against black rot:

- Contact agents should be applied before infection, although they may not be fully effective. Unfortunately, organic growers are still in a difficult position, because the sulphur and copper they commonly use are virtually ineffective against black rot. Dithiocarbamates are generally considered to have excellent efficacy against black rot, although there are conflicting experiences from farmers. Unfortunately, fungicides containing mancozeb and myclobutanil have already been withdrawn from commercial distribution in the EU. Metiram, phthalimide fungicides such as captan and folpet, dithianon and fluxapyroxad are still approved, but other new plant protection products containing active substances such as the combination of fluopyram + tebuconazole and the recently registered mefentrifluconazole are also permitted in the EU [106].
- Fungal sterol biosynthesis inhibitors: difenoconazole, flusilazole, tetraconazole and tebuconazole have excellent activity against powdery mildew, and many of them are

also very effective against black rot [49]. Their preventive effect is weak, but they can block the disease process through their rapid absorption, stopping the development of symptoms in both leaves and berries. The long-lasting curative effect of the withdrawn myclobutanil and perhaps the equally effective triazole fungicides, tebuconazoles, provides flexibility in the timing of black rot control. They should therefore be used mainly after infection, in the first half of the incubation period of *G. bidwellii*. Their weak preventive effect can be improved by combining them with the still available dithiocarbamate agents.

- Of the strobilurins, pyraclostrobin is the best preventive and curative agent for both leaves and berries [107]. This active ingredient, sprayed after flower caps have fallen off, binds well to the waxy layer and follows surface growth of black rot on leaves and berries; therefore, has a long duration of action of up to 3 weeks. Although the efficacy of azoxystrobin and kresoxim-methyl has been confirmed [81,108], the excellent efficacy of pyraclostrobin unfortunately does not automatically apply to other strobilurin-type agents such as azoxystrobin [107]. In any case, it is reassuring that the development of strobilurin resistance in black rot is not a concern [109].

It should be noted that, according to the latest compilation by the Fungicide Resistance Action Committee [110], *G. bidwellii* has no known resistance to any fungicide, but it is expected that its recent re-emergence in vineyards could result in the selection of *G. bidwellii* strains that are less sensitive or even resistant to fungicides belonging to certain chemical families. Therefore, alternate use of fungicides with different modes of action is recommended [111].

### 7.6. On the Road to Sustainability

Currently, pesticides are the most effective means of controlling several plant pathogens, but to move towards sustainability, this dependence should be reduced, not only through efficiency and substitution, but also through a complete restructure of familiar crop protection technologies. Crop protection practices such as mechanical techniques, plant breeding, biological control, induced resistance, the application of ecological principles, precision agriculture and new pesticides can all contribute to the development of sustainable agricultural systems and have the greatest impact when used together in the most appropriate combinations [112].

In addition to the redesign and development of practical crop production and crop protection technologies, research in the biological sciences also has an important role to play in achieving sustainability goals. Biologists and plant pathologists have a huge task in exploring the myriad interactions between plants and their environment and between plants and different microscopic and macroscopic organisms that together influence plant health and productivity.

## 8. Metabolite Analyses Associated with Black Rot—Grapevine Interactions

### 8.1. Secondary Metabolite Identifications

In order to elaborate efficient disease-management strategies, a better understanding of host-pathogen interactions, particularly pathogenicity (the potential to cause disease) and virulence (the disease-causing power) of phytopathogenic fungi, such as *G. bidwellii*, are of great importance. Pathogenicity is a qualitative term, the quality of being pathogenic, while virulence quantifies disease-producing properties in a given strain of a pathogenic species [113,114]. Secreting appropriate virulence factors (hydrolytic enzymes, effector proteins, and low molecular weight substances), plant pathogens can invade host plants and to multiply and establish there [115–119]. Fungi, including phytopathogens, produce a number of structurally diverse low molecular weight substances (secondary metabolites) [120] in various strategies of fungal survival (e.g., antibiosis, antagonism, competition, parasitism, symbiosis) [121]. Among them, phytotoxins are determinant virulence factors in the multifaceted process of plant-pathogen interactions [122].

Compared to other plant pathogens [123–128], few studies have so far addressed characterization of virulence factors of *G. bidwellii* (Table 2), and only two recent studies have attempted to investigate the impact of *G. bidwellii* infection in grapevine [129,130]. In order to support the hypothesis that the necrotic lesions on infected plant organs after infection with *G. bidwellii* are an indication of phytotoxins secreted by the fungus [50], and to gain deeper insights into the interaction of *G. bidwellii* and *V. vinifera*, Buckel et al. [131] were the first to make *in planta* experiments. However, it should be noted that they performed experiments only on the fungal side, and the *G. bidwellii* strain used in their study (CBS 111645) has been recognized as *Phyllosticta parthenocissi* rather than *Phyllosticta ampelicida* (anamorph of *G. bidwellii*) since 2013 [22]. Nevertheless, they used infected and non-infected leaf material of *V. vinifera* to prove whether phytotoxic dioxolanone-type secondary metabolites (Figure 4) are present in the pathogenic process. Using HPLC-MS, a signal was found on the chromatogram of the infected sample that was not present in the uninfected sample. The signal was identified as a phytotoxic secondary metabolite: guignardic acid (Table 2). This dioxolanone ring-containing compound was first discovered and characterized by Rodrigues-Heerklotz et al. [132]. It was detected from the fermentation broth of a selected *Guignardia* strain (anamorph name: *Phyllosticta telopeae*), the crude extract of which demonstrated previously some degree of antibacterial activity. *P. telopeae* was found on a native Australian plant species, *Telopea speciosissima* [133]. It is not a major plant-pathogen, rather described as endophytic fungus [132].

Table 2. Fungal secondary metabolites associated with grape black rot, published until 2022.

| Secondary Metabolite | Compound Summary Active PubChem Weblinks | Compound Phytotoxicity | Producing Phyllosticta Strains, Host Plants and Fungal Lifestyle | | | Mode of Compound Detection | |
|---|---|---|---|---|---|---|---|
| | | | | | | Submerged Culture Fermentation | *In planta V. vinifera*—Fungus Interaction |
| Guignardic acid | https://pubchem.ncbi.nlm.nih.gov/compound/Guignardic-acid (accessed on 30 September 2022) | Highly phytotoxic, not host-specific | *P. telopeae* | *Telopea speciosissima* | Endophyte | [132] | NDA |
| | | | *P. parthenocissi* CBS 111645 | *Parthenocissus quinquefolia* | Plant pathogen | [134] | [131,135] |
| | | | *P. musarum* IMI 147360 | *Musa* sp. | Plant pathogen | [131] | NDA |
| | | | *P. elongata* CBS 126.22 | *Vaccinium* sp. | Casual plant pathogen | [131] | NDA |
| | | | *P. capitalensis* CBS 123405 | Recorded on 70 plant families * | Endophyte and plant pathogen | [131] | NDA |
| Phenguignardic acid | https://pubchem.ncbi.nlm.nih.gov/compound/72204238 (accessed on 30 September 2022) | Highly phytotoxic, not host-specific | *P. parthenocissi* CBS 111645 | *Parthenocissus quinquefolia* | Plant pathogen | [134] | NDA |
| | | | *P. sphaeropsoidea* CBS 756.70 | *Aesculus* sp. | Plant pathogen | [131] | NDA |
| | | | *P. musarum* IMI 147360 | *Musa* sp. | Plant pathogen | [131] | NDA |
| | | | *P. elongata* CBS 126.22 | *Vaccinium* sp. | Casual plant pathogen | [131] | NDA |
| Alaguignardic acid | https://pubchem.ncbi.nlm.nih.gov/compound/102435402 (accessed on 30 September 2022) | Highly phytotoxic, not host-specific | | | | | NDA |
| Guignardianone A | https://pubchem.ncbi.nlm.nih.gov/compound/102435403 (accessed on 30 September 2022) | Phytotoxic, not host-specific | | | | | NDA |
| Guignardianone B | https://pubchem.ncbi.nlm.nih.gov/compound/102435404 (accessed on 30 September 2022) | Non-phytotoxic | | | | | NDA |
| Guignardianone C | https://pubchem.ncbi.nlm.nih.gov/compound/102435405 (accessed on 30 September 2022) | Non-phytotoxic | *P. parthenocissi* CBS 111645 | *Parthenocissus quinquefolia* | Plant pathogen | [136] | NDA |
| Guignardianone D | https://pubchem.ncbi.nlm.nih.gov/compound/102435406 (accessed on 30 September 2022) | Non-phytotoxic | | | | | NDA |
| Guignardianone E | https://pubchem.ncbi.nlm.nih.gov/compound/71682251 (accessed on 30 September 2022) | Phytotoxic, not host-specific | | | | | NDA |
| Guignardianone F | https://pubchem.ncbi.nlm.nih.gov/compound/71682252 (accessed on 30 September 2022) | Phytotoxic, not host-specific | | | | | NDA |
| (6S,9R)-vomifoliol | https://pubchem.ncbi.nlm.nih.gov/compound/5280462 (accessed on 30 September 2022) | | | | | | NDA |
| Guignarenone A | NDA | | | | | | NDA |
| Guignarenone B | NDA | Non-phytotoxic | *P. ampelicida* PSU-G11 | *Garcinia hombroniana* | Endophyte | [137] | NDA |
| Guignarenone C | https://pubchem.ncbi.nlm.nih.gov/compound/139585121 (accessed on 30 September 2022) | | | | | | NDA |
| Guignarenone D | NDA | | | | | | NDA |

* [138].

**Figure 4.** 2D structures of phytotoxic secondary metabolites of grape black rot discovered to date [139–144].

Before the *in planta* experiments of Buckel et al. [131], using the same CBS 111645 strain, but from submerged culture [134], guignardic acid and another phytotoxic compound—phenguignardic acid (Table 2)—were detected, and it was found that both compounds are highly phytotoxic and not host specific in plant experiments on grapevine, wheat and rice. However, grapevine leaves were more sensitive to these compounds than the other non-host plant leaves and phenguignardic acid was five-ten times more phytotoxic then guignardic acid. Furthermore, Buckel et al. [131] demonstrated that phenguignardic acid is biosynthetically derived from two molecules of phenylalanine (Figure 4), but strikingly, in their experiment, while guignardic acid was induced, phenguignardic acid was not detected in real interactions with grapes.

Previously, using optimized submerged culture of the above-mentioned *Phyllosticta* CBS 111645 strain, which was classified as *G. bidwellii* that time, Buckel et al. [136] found seven additional secondary metabolites that are structurally related to guignardic acid. Four of the seven compounds, alaguignardic acid and the guignardianones A, E and F, showed phytotoxic activities (Table 2; Figure 4); furthermore, alaguignardic acid exhibited comparable phytotoxic potential to guignardic acid. Interestingly, none of them was detected later *in planta* in the experiment of Buckel et al. [131] but it was confirmed that phenylalanine is a key precursor in the biosynthesis of alaguignardic acid and guignardic acid.

It is important to point out that the fungal strain CBS 111645 used in most of the studies was not *Phyllosticta ampelicida* (anamorph of *G. bidwellii*) but *Phyllosticta parthenocissi*, i.e., a fungal strain from the same genus but of a different species. As *P. parthenocissi* is a near relative of *P. ampelicida,* they have likely comparable metabolite profile in similar plant-pathogen interactions. Indeed, Buckel et al. [131] found that four other closely related *Guignardia* species (from which three was plant pathogenic and one endophytic) produced the same compounds. All the four species produced one or more phytotoxic dioxolanones in submerged cultures, known from CBS 111645 strain (Table 2): *Guignardia aesculi* CBS 756.70 (anamorph: *Phyllosticta sphaeropsoidea*), that causes leaf blotch

disease on *Aesculus* sp. produced phenguignardic acid. Banana freckle disease agent *Guignardia musae* IMI 147360 (anamorph: *Phyllosticta musarum*), and *Guignardia vaccinia* CBS 126.22 (anamorph: *Phyllosticta elongata*) the pathogen infecting *Vaccinium* sp. produced both guignardic acid and phenguignardic acid. *Guignardia mangiferae* CBS 123405 (anamorph: *Phyllosticta capitalensis*), which was considered by Buckel et al. [131] as endophyte, also produced guignardic acid.

*P. capitalensis* is indeed isolated worldwide as an endophyte, although it was originally described by Hennings [145] as a fungal pathogen of *Stanhopea* (*Orchidaceae*) in Brazil, and is now considered in the scientific literature not only as an endophyte but also as a plant pathogen of minor importance [55,138]. Furthermore, the concept of "endophytic fungi" is not consistently used in the scientific literature and this is likely to be the case for *P. capitalensis*. According to some authors, endophytic fungi reside inside host tissues for all or part of their life cycle without causing visible symptoms on the host plant [42], while other authors argue that endophytic fungi can potentially become saprotrophs or pathogens, usually when the plant is stressed [135,146]. Thus, it is not entirely surprising that the strain CBS 123405 of endophytic *P. capitalensis* also produced some level of phytotoxic guignardic acid in the experiments of Buckel et al. [131].

It is also worth mentioning, that from a non-pathogenic endophyte *G. bidwellii* strain PSU-G11, which was isolated from leaves of *Garcinia hombroniana* and identified by ITS1-5.8S-ITS2 regions of its ribosomal RNA gene because it did not produce any conidia or spores, additional secondary metabolites were isolated by Sommart et al. [137]: four new tricycloalternarene derivative guignarenones A–D, along with known (6*S*,9*R*)-vomifoliol [147] (Table 2). These newly characterized compounds and (6*S*,9*R*)-vomifoliol have neither known phytotoxic activity nor a proven role in the interaction between black rot and grapevine. Although, Sommart et al. [137] found that guignarenones A had mild cytotoxic activity against oral cavity cancer and African green monkey kidney fibroblast (Vero) cell lines, and the glucoside of (6*S*,9*R*)-vomifoliol, roseoside, and its related compounds were previously described as inhibitors of histamine release from rat peritoneal exudate cells induced by antigen-antibody reaction [148].

Phytotoxins are products of plant pathogens during host-pathogen interaction that directly injure plant cells, and influence the course of disease development and symptoms. Both fungal and bacterial pathogens produce a number of secondary metabolites that are toxic to plant cells; however, these metabolites may not be important in plant disease. Consequently, phytopathologists have developed criteria for assessing the involvement of toxins in plant disease [149]. These include (i) reproduction of disease symptoms with the purified toxin, (ii) a correlation between toxin yield and pathogenicity, (iii) production of the toxin during active growth of the pathogen *in planta*, and (iv) reduced virulence or lack of virulence in nontoxigenic strains. So far, some of the above-mentioned dioxolanones (Figure 4) have been shown to have phytotoxic effects (Table 2), but none of them met all four criteria for true phytotoxins. In the experiments of Molitor et al. [134] and Buckel et al. [136], the observation of phytotoxic lesions on leaves of grapes, rice and wheat due to the application of guignardic-, phenguignardic-, alaguignardic acid and guignardianones A, E and F to plant leaves confirms the first criterion. Likewise, a correlation between the amount of toxin and pathogenicity was observed for guignardic- and phenguignardic acids, so that the second criterion was met for at least these two compounds [134]. As for the third criterion, only the production of guignardic acid has been proven *in planta* during pathogen-plant interaction [131]. Furthermore, avirulent endophytic *G. mangiferae* was a potential species to demonstrate the fourth criterion, but the results obtained by Buckel [150] did not provide a definitive answer as the fungus also produced guignardic acid. Thus, none of the above toxins fulfilled all four criteria, and their role in the development of plant disease, including black rot of *V. vinifera*, has not yet been confirmed.

The majority of these scientific works were not based on real interactions between black rot and grapes, with the exception of Buckel [150] and Buckel et al. [131], but their inquiries were exclusively focused on the metabolites produced by the fungus. If not to

explore the complex bilateral molecular interaction between black rot and grapes, but to investigate the consequence of their interactions, three other scientific publications have been published. Two of them were carried out from an oenological point of view and focused on the impact of the infection on the intrinsic value of the berries and on certain metabolites produced by the grape [129,130].

Plant pathogens, such as *G. bidwellii*, can have an indirect impact on wine quality by altering metabolic pathways and producing compounds with off-flavours and/or aromas in grapevine berries. Kellner et al. [129] were the first to examine selected black rot-affected berries for compounds that, when added to wine in above-average amounts, may reduce the organoleptic value of the wine, or even pose a health risk to consumers. The examined secondary metabolites of grapevine, such as polyphenols, as one of the most important groups of compounds from oenological aspect, and resveratrol content did not change significantly compared to healthy berries, but a significant accumulation of sugars and high levels of succinic, fumaric and caftaric acids were measured, indicating abnormal berry-metabolism. Further analysis showed that, in all examined grape varieties, the targeted mycotoxins and fungal biogenic amines were absent or were present only in low concentrations (e.g., ochratoxin A and histamine). However, it is unclear whether these compounds were actually produced by *G. bidwellii* or by other opportunistic fungal species, as the authors themselves note that the damage caused by active penetration of *G. bidwellii* can open the berry surface to other toxin-producing fungi such as *Penicillium* and *Aspergillus* species. Finally, Kellner et al. [129] concluded that it is advisable to remove black rot-infected berries before vinification for quality reasons, but their accidental presence in must does not pose any health problem.

In another oenological study, Poitou et al. [130] detected methyl salicylate at the highest concentrations in wines of several grape varieties, made from grapes affected by Esca, *Plasmopara viticola* and *G. bidwellii*. Comparing healthy, sun shrivelled and fungi infected Merlot berries; *G. bidwellii* induced the highest concentration of methyl salicylate (Figure 5).

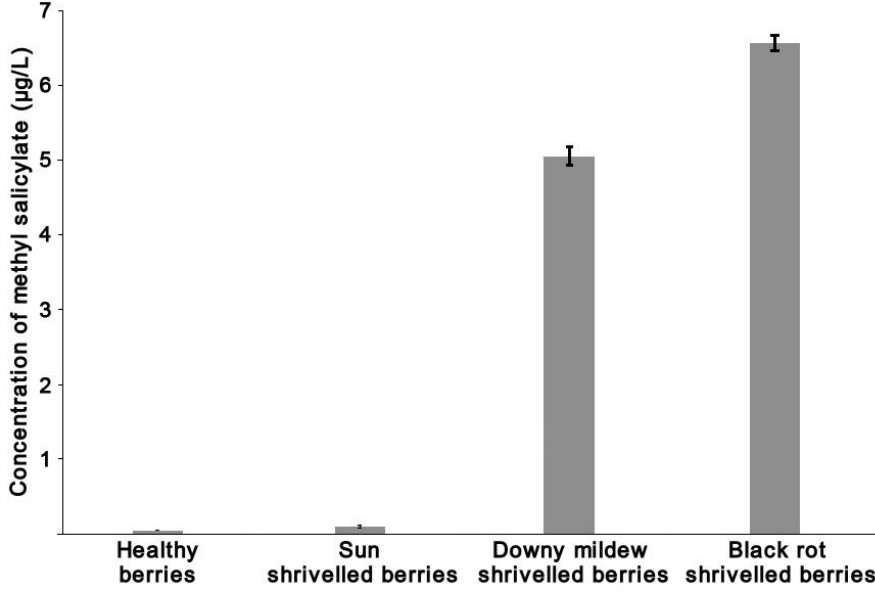

**Figure 5.** Concentration of methyl salicylate (µg/L) in fermented model must with Merlot berries: healthy control, sun-shrivelled berries and shrivelled berries affected by downy mildew and grape black rot [130]. Reprinted from Food Chemistry, 360, Poitou, X.; Redon, P.; Pons, A.; Bruez, E.; Delière, L.; Marchal, A.; Cholet, C.; Geny-Denis, L.; Darriet, P. Methyl Salicylate, a Grape and Wine Chemical Marker and Sensory Contributor in Wines Elaborated from Grapes Affected or Not by Cryptogamic Diseases, 130120, Copyright (2021), with permission from Elsevier.

Methyl salicylate is a volatile odorous secondary plant metabolite, a benzoate ester, a member of salicylates and produced by many species of plants, particularly wintergreens. The compound was extracted and isolated from plant species *Gaultheria procumbens* in 1843 and frequently used as a fragrance, in foods, beverages, and liniments [151]. It is implicated in a number of phytopathological processes, such as the induction of systemic acquired stress in diseased plants [152]; resistance activation of neighbouring plants through its releasement into the air [153]; direct insecticidal effect [154]; attracting effect for natural insect predators against agricultural pest insects [155]. It has been also evidenced as a biomarker for downy mildew infected grapevine leaves [156], but until the work of Poitou et al. [130], methyl salicylate had not been associated with black rot of grapes. In the study of Poitou et al. [130], it has been concluded, that the change of methyl salicylate concentration was induced by host defence mechanism against fungal infection and the vinification of such diseased grapes may affect the wine aroma quality. Furthermore, methyl salicylate can be a good indicator of the infestation status of vineyards, revealing the physiological state of vine plants under the pressure of fungal infection.

As part of a larger project to explore the utilization of *V. sylvestris* for crossbreeding to obtain resistant grapevine varieties against downy mildew, powdery mildew and black rot [157], an extensive study was carried out on the biology of *G. bidwellii* and the defence mechanisms of different *V. sylvestris* genotypes, rootstock cultivar 'Börner', and cultivated vine varieties 'Solaris' and 'Müller-Thurgau' against *G. bidwellii* [46]. In addition to the valuable in vitro nutritional studies of black rot, the in vivo histological observations of interactions between black rot and grapevine, and the determination of constitutively present phenolic compounds in different grapevine genotypes, transcriptomic and metabolomic investigations of the pathological process have also been carried out, targeting various defence-associated genes and secondary metabolites. In terms of secondary metabolites, the focus was on phenolic compounds; as these have repeatedly been shown to have inhibitory effects against fungal pathogens [158–160]. For this purpose, whole plants of 'Müller-Thurgau', 'Solaris' and 'Börner' were sprayed with black rot conidia suspension on the upper and lower surfaces of the leaves and following infection both free and cell wall-bound phenols were determined by Folin-Ciocalteu assay and HPLC. Interestingly, infection-related differences in phenol concentration were hardly observed, with only 'Solaris' showing a significant difference between the infected and the corresponding mock-inoculated variant 2 dpi. In summary, the author suggests that sampling time points, inoculation and detection methods should be reconsidered before further studies.

In the same study, quantitative PCR was used to investigate the transcript dynamics of various defence-associated genes in different cultivars after inoculation with *G. bidwellii*. These in vitro experiments were based on excised leaf discs (Ø = 14 mm) inoculated with droplets of a conidial suspension of black rot. As a result, increased transcript levels of phenylalanine ammonia lyase, a key enzyme of phenol metabolism, were determined in both the resistant 'Börner' and the partially resistant 'Solaris' cultivars compared to the susceptible 'Müller-Thurgau' cultivar. A very similar picture was also seen in the transcript profiles of stilbene synthases, which are also involved in phenol metabolism. The authors note, that increased induction of phenylalanine ammonia lyase and other enzymes related to secondary metabolism was also found after infections with downy mildew in the resistant grapevine species *V. riparia*, which is a parent of the cultivar 'Börner' [161].

In the same project [157], Duan et al. [162] found that among 86 genotypes of *V. sylvestris* those that produce high levels of stilbene (piceid, resveratrol, piceatannol pterostilbene and viniferin) in response to UV-C are significantly less susceptible to downy mildew infection than genotypes with low UV-C inducibility of stilbene. Using promoter reporter assays, it was also found that in a *V. sylvestris* genotype (Hö29), the strong induction of stilbenes by UV light and downy mildew is due to a fragment of the *MYB14* transcription factor promoter that is absent in *V. vinifera* [163], making it a potential target for resistance breeding.

These results suggest that phenolic compounds may have an influence on the expression of resistance to black rot in certain grapevine cultivars, and genotypes of *V. sylvestris* that produce high levels of stilbenes can provide valuable genetic resources for marker-assisted breeding to improve basal immunity. However, *in planta* detection and quantification of individual polyphenols, particularly certain stilbenes, in the mechanisms of grapevine defence against black rot still needs to be performed.

*8.2. Metabolomics, a Promising Tool to Describe the Grapevine—Black Rot Pathosystem*

Molecular characterization of black rot—grapevine interaction has rarely been attempted. Most of the scientific works published to date and discussed in this review have focused on the finding and characterization of toxic secondary metabolites, mostly seeking fungal compounds useful for agricultural and medicinal purposes. To induce and extract these compounds, submerged fermentation systems without plant partners were used almost exclusively. However, characterization of microbial virulence factors and inducible plant defence mechanisms involved in the complex interaction between host plant and plant pathogen—that is, elucidation of primary and secondary metabolites during actual pathogenesis—promises to reveal many more compounds and enzymes beneficial for humans. With the development of holistic and integrative "systems biology" approaches and "omics" technologies, which systems biology brings such as proteomics, genomics, transcriptomics, metabolomics and phenomics, it is becoming increasingly possible to understand interactions between host plants and plant pathogens [164,165].

Metabolomics reveals the metabolite profile of a biological system; its different chemical compositions, physiology and complexity at a given time point [166]. Among the "omics," it has recently evolved into a powerful approach for exploring the role of metabolites in plant-microbe interactions in non-model plants, since metabolites reflect the end point of biological activities and their levels can be regarded as the ultimate response of a biological system to genetic or environmental changes [167,168]. During plant-pathogen interactions, metabolomics can be used to detect metabolites that have a role in pathogen surveillance, signal transduction, enzyme regulation, cell-to-cell signalling, and antimicrobial activity in plants [169], and also molecules secreted by pathogens during colonization [170], or amino acids and sugars whose production is induced or mislocalized in the host to promote pathogen growth [171]. In addition to identifying new metabolites, the integration of metabolomics data with the other "omics" also allows the elucidation of major metabolic pathways, biological processes that control metabolite levels and complex metabolic networks during plant–pathogen interactions [172,173].

On the other side, metabolomics has also potential limitations such as sensitivity to external influences such as temperature, light levels, nutritional status etc., although, by carefully constructed and monitored experiments, and the results interpreted with respect to the experimental conditions, metabolomics holds great potential [174]. Another challenge in the study of plant host-pathogen interactions is the distinction between host plant and pathogen metabolites. A possible solution is molecule labelling. For example, Pang et al. [175] successfully labelled metabolites from *Pseudomonas syringae* using heavy isotopes and distinguished them from metabolites from *Arabidopsis* stomatal cells, allowing the identification of bacterial amino and organic acids. Another difficulty is the identification of unknown compounds, because there are no specific metabolite databases for plants and plant pathogens, such as those used for human and yeast studies [171]. Nevertheless, advanced analytical tools, like gas chromatography-mass spectrometry (GC-MS), liquid chromatography mass-spectroscopy (LC-MS), capillary electrophoresis-mass spectrometry (CE-MS), Fourier transform ion cyclotron resonance-mass spectrometry (FTICR-MS) matrix-assisted laser desorption/ionization (MALDI), ion mobility spectrometry (IMS) and nuclear magnetic resonance (NMR) are speeding up precise natural product discovery. Misra et al. [176] and Patel et al. [177] provide excellent overviews of these various techniques, procedures, analytical and bioinformatics tools, and plant metabolomics databases, the latter two of which will expectedly become increasingly user-friendly for researchers.

In summary, the resistance or susceptibility of *V. vinifera* to black rot disease depends on the characteristics of both the plant and the fungus concerned. Advances in metabolomics offer promising approaches to deciphering the multifaceted physiological and biochemical processes behind compatible or incompatible pathogen-plant interaction and may also contribute to the discovery of compounds with potential practical applications (e.g., bio-herbicides, bio-fungicides and medicines). Furthermore, with the discovery of secondary metabolite biomarkers associated with resistance of black rot, metabolomics-assisted breeding may allow efficient and rapid phenotyping of grapevine seedlings with inherited resistance immediately after germination, thus considerably reducing costs and significantly shortening breeding procedures of grapevine for black rot resistance [178–180].

**Author Contributions:** Conceptualization, M.S. and T.D. (Tamás Deák); writing—original draft preparation, M.S., A.C.-K., T.D. (Terézia Dula), D.R. and T.D. (Tamás Deák); writing—review and editing, M.S., A.C.-K., E.F., P.K. and T.D. (Tamás Deák); visualization, M.S. and A.C.-K.; supervision, P.K.; funding acquisition, T.D. (Tamás Deák) and P.K. All authors have read and agreed to the published version of the manuscript.

**Funding:** This research (no. K125476) has been implemented with the support provided by the Ministry of Innovation and Technology of Hungary from the National Research, Development and Innovation Fund, financed under the K_17 funding scheme.

**Data Availability Statement:** Not applicable.

**Conflicts of Interest:** The authors declare no conflict of interest.

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
