# Peer review of "Black Rot of Grapes (Guignardia bidwellii)—A Comprehensive Overview"

_horticulturae, doi:10.3390/horticulturae9020130_

Round 1
Reviewer 1 Report
check the names of chemical substances for instance Strobilurines is written wrong at least once
Author Response
We are grateful for the positive opinion. English has been carefully edited once more by an outside lector. The typo in "strobilurines" has been corrected and we double checked spelling of chemical substances once more.
Reviewer 2 Report
The article is well written. Some minor comments
-Always a sentence should be started with the full scientific name
-anamorph and teleomorph need to be changed to asexual and sexual morph

Author Response
Thank You for the positive opinion which helped to improove the manuscript.
Note1: Always a sentence should be started with the full scientific name
Thank You for pointing this out. During language editing, all instances where the sentence started iwth a latin name, the genus is written out and not abbreviated.
Note2: -anamorph and teleomorph need to be changed to asexual and sexual morph
We replaced anamorph and teleomorph to asexual stage and sexual stage.
Reviewer 3 Report
Concerning the paper „Black rot of grapes (Guignardia bidwellii) – A comprehensive overview” on pruning, I think it is an useful paper that illustrates up dated pathogen knowledge , its epidemiology and the management of disease (Minor revision). It will be a valuable source of information for science audience who works on this issue.
Some details that should be corrected is concering black rot as disease and Guignardia bidwellii as pathogen , namely authors should correct some parts in the text e.g. line 122 „Black rot of grapes is a typical polycyclic pathogen ….”; black rot is not a pathogen it is disease, please correct or rewritte thorough the whole text.
Author Response
Thank You very much for the positive review. Your remark ("Some details that should be corrected is concering black rot as disease and Guignardia bidwellii as pathogen , namely authors should correct some parts in the text e.g. line 122 „Black rot of grapes is a typical polycyclic pathogen ….”; black rot is not a pathogen it is disease, please correct or rewritte thorough the whole text.") is completely valid, thank you for pointing this out. We carefully screened the document and named the disease where the disease is discussed and the pathogen where the pathogen was discussed.
Reviewer 4 Report
I think the authors should be congratulated for developing this outstanding and informative review as well as a great job of reviewing the manuscript.
I
Author Response
Thank you for the positive and encouraging opinion. We hope that the revised manuscript has improved further.
Reviewer 5 Report
The manuscript needs editing and improvment in reducing redundancy and eliminating them. The appropriate wording is also necessary as specified in several points in the partly annotated version of the manuscript enclosed.

Author Response
Thank you for the valuable review and the suggestions. The responses are to be found in the attached file.
We submitted the manuscript for language editing and reduced and simplified specific parts.

Round 2
Reviewer 5 Report
A few revisions listed in the enclosed annoated manuscript as needed.

Author Response
Thank You for the review, we applied the required changes. Please see the attached file.
